# Pedestrian Navigation Method Based on Machine Learning and Gait Feature Assistance

**DOI:** 10.3390/s20051530

**Published:** 2020-03-10

**Authors:** Zijun Zhou, Shuqin Yang, Zhisen Ni, Weixing Qian, Cuihong Gu, Zekun Cao

**Affiliations:** 1School of Electrical and Automation Engineering, Nanjing Normal University, Nanjing 210023, China; a15050528624@163.com (Z.Z.); 15651785167@163.com (S.Y.); nizhisen@stu.njnu.edu.cn (Z.N.); 171835008@stu.njnu.edu.cn (C.G.); 171835001@stu.njnu.edu.cn (Z.C.); 2College of Automation Engineering, Nanjing University of Aeronautics and Astronautics, Nanjing 210016, China

**Keywords:** pedestrian navigation, virtual inertial navigation system, machine learning, gait phase recognition, gait feature assistance

## Abstract

In recent years, as the mechanical structure of humanoid robots increasingly resembles the human form, research on pedestrian navigation technology has become of great significance for the development of humanoid robot navigation systems. To solve the problem that the wearable inertial navigation system based on micro-inertial measurement units (MIMUs) installed on feet cannot effectively realize its positioning function when the body movement is too drastic to be measured correctly by commercial grade inertial sensors, a pedestrian navigation method based on construction of a virtual inertial measurement unit (VIMU) and gait feature assistance is proposed. The inertial data from different positions of pedestrians’ lower limbs are collected synchronously via actual IMUs as training samples. The nonlinear mapping relationship between inertial information from the human foot and leg is established by a visual geometry group-long short term memory (VGG-LSTM) neural network model, based on which the foot VIMU and virtual inertial navigation system (VINS) are constructed. The VINS experimental results show that, combined with zero-velocity update (ZUPT), the integrated method of error modification proposed in this paper can effectively reduce the accumulation of positioning errors in situations where the gait type exceeds the measurement range of the inertial sensors. The positioning performance of the proposed method is more accurate and stable in complex gait types than that merely using ZUPT.

## 1. Introduction

Pedestrian navigation systems, with their light weight, small size, cheap price, and convenience to carry, show wide prospects in military and civilian application. Pedestrian navigation technology has become a hot issue in recent years as an important branch of navigation technology [1]. In the case of the outdoors, the location information can be obtained by using the global navigation satellite system (GNSS), as well as other precision-navigation technologies that fuse GNSS and sensor data with map information [2]. Although these technologies are relatively mature, they need indoor and outdoor deployment of related devices, and are vulnerable to interference from the surrounding environment [3]. Therefore, scholars have begun to study the autonomous pedestrian navigation methods in view of the insufficiency of satellite-based pedestrian navigation technology.

Wearable pedestrian navigation system navigation sensors refer to micro-electro-mechanical system (MEMS) inertial sensors, magnetometers, barometric altimeter and so on [4]. With the rapid development of MEMS technology, MEMS inertial sensor-based pedestrian navigation systems play an important role in personal positioning indoors and outdoors [5,6,7,8]. According to the installation positions of micro-electro-mechanical system-inertial measurement units (MEMS-IMUs), pedestrian navigation systems can be classified into pedestrian dead reckoning (PDR) systems and strapdown inertial navigation systems (SINS) assisted by zero-velocity update (ZUPT) technology. The PDR algorithm needs to conduct kinematic modeling of human body, calculate the step length according to parameters such as step frequency and leg length, and obtain the 3D pedestrian position information with the aid of magnetic sensors. However, the parameters of different individual models vary greatly and PDR theories also have limitations when dealing with complex gait types [9,10]. Strapdown inertial navigation methods assisted by ZUPT use accelerometers and gyroscopes to calculate the navigation parameters of the human feet by a SINS algorithm, and ZUPT algorithm is used to suppress the accumulation of navigation errors when human feet are in static gait phases. The method is based on the fact that when a MEMS-IMU is installed on the foot of a pedestrian, the zero-velocity at the moment of the foot being on the ground is taken as an observation, and the zero-velocity and SINS information are fused to obtain the modified information of the MEMS-SINS, and sequentially to improve the accuracy of the pedestrian navigation system [11,12,13].

In terms of the previous pedestrian navigation systems based on wearable navigation sensors, Klingbeil et al. [14] used a foot-bound strapdown navigation algorithm, with the walking distance of the pedestrian being obtained by the velocity integral of the foot in the swinging phase. Moreover, a gyroscope threshold was designed to detect whether the foot was in the swinging phase. Alvarez et al. [15] presented a waist-worn personal navigation method based on IMUs, and described an improved algorithm based on a detailed description of the heel strike biomechanics, as well as its translation to acceleration of the body waist to estimate the periods of zero velocity, the step length, and the heading of the pedestrian. Liang et al. [16] presented a wearable inertial pedestrian navigation system and its associated pedestrian trajectory reconstruction algorithm. They utilized a sensor fusion technique based on a double-stage quaternion-based extended Kalman filter to fuse acceleration, angular velocity, and magnetic signals. To sum up, although the required performance in stationary motions can be achieved by current pedestrian navigation technologies, it may be affected by certain adverse factors, such as continuous over-range and fault of inertial sensors while the foot inertial pedestrian navigation system works under the circumstances of complex and strenuous human motions.

In relevant research fields, neural network (NN) models are essentially statistical models which can establish a relationship or mapping between input and output through learning algorithms without the need for determining the model of the system. Due to this characteristic, the application of neural networks in navigation has become a research hot spot and trend. Aboelmagd [17] proposed a global positioning system/ inertial navigation systems (GPS/INS) integrated navigation technology based on input delay neural network (IDNN), and the results showed that IDNN method showed better navigation performance than the GPS/INS integrated navigation system based on Kalman filter when GPS signals were occluded or shielded. Chiang et al. [18] studied and developed an embedded sensor fusion algorithm based on a cascade correlation neural network, which could automatically determine the prediction task with a simpler, more flexible topology structure and less training workload, thus improving the accuracy of positioning parameters during GPS downtime. Ko et al. [19] verified the potential of neural network-based autonomous navigation for mobile homecare. The result suggested that the recurrent neural network can perform better robot navigation because of its capability to handle the temporal dependency of a data sequence.

Inspired by the research works on NN mentioned above, a novel pedestrian navigation method based on gait feature assistance and construction of a virtual foot IMU is proposed. It can be learned from previous experience that inertial sensors can be installed on the human hand, waist, shoulders and other parts, where there are periodic changes during walking. Due to the large acceleration of the foot, it is convenient to make use of the inertial information at rest to correct it. Therefore, the method proposed in this paper chooses to bind the IMUs to thigh and foot [20]. Human walking is a regular, cyclical complex movement, in which the inertial information between pedestrian’s thigh and foot follows the nonlinear kinematic model of the human body. The lever arm effect between thigh and foot can be eliminated by the use of kinematic algorithms [21], as well as by neural networks for regression [22]. Because the characteristics of a non-rigid body make the human rigid model inaccurate, a machine learning method is used in this paper to establish a nonlinear model of the relationship between the thigh and foot inertial information. Since the visual geometry group-long short term memory (VGG-LSTM) neural network model can take advantage of the structural characteristics of hierarchical timing sequence of inertial sensor data to conduct more comprehensive data mining, it can be used to construct a virtual foot IMU from an actual thigh IMU according to the kinematic model of human body and complete the virtual inertial navigation system. Furthermore, the attitude information from the virtual inertial navigation system (VINS) is modified via the repeatability feature of human walking motion. With the aid of the ZUPT algorithm, the accumulation of positioning error along with the travel distance can be further slowed down to meet the requirement of both indoor and outdoor application.

In the next section, an overview and design basis of the pedestrian navigation method is introduced. Section 3 presents the construction of virtual inertial measurement unit (VIMU) based on machine learning. Section 4 provides the pedestrian navigation algorithm with gait feature assistance. Experimental results from the pedestrian navigation system are presented and analyzed in Section 5. Finally, Section 6 concludes the paper.

## 2. System Overview

Figure 1 shows the process of the proposed pedestrian navigation method, which consists of two parts. Part 1 is the construction process of neural network model, which is completed in advance and is not included in the actual pedestrian navigation process. Part 2 is the actual pedestrian navigation process. Both parts will be introduced in detail in this section.

### 2.1. Part 1

The foot acts as the key position for human motion information collection in pedestrian navigation systems based on wearable sensors. However, MIMUs mounted on feet may exceed their measurement range or malfunction when the human motion range includes strenuous actions. Aiming at resolving this problem, this paper presents a virtual foot IMU construction method based on machine learning. 

As shown in Figure 1a, in order to construct the virtual foot IMU, a neural network model (VGG-LSTM) is needed to realize the nonlinear mapping function between the inertial information of foot and leg. In order to construct the VGG-LSTM neural network model, MEMS IMUs should be installed on the leg and foot of the same side of lower limb, respectively, and the tri-axial acceleration and tri-axial angular velocity of the IMUs collected at the same frequency are used as the training samples of the neural network model. When the model is completed after training, testing and verifying, the virtual foot IMU can be constructed from the inertial information of the leg. The details of neural network model will be illustrated in Section 3.

### 2.2. Part 2

As shown in Figure 1b, when the pedestrian is performing a strenuous action while the MEMS IMU is installed on the leg, the neural network model constructed in advance can be used as the approximation function of the nonlinear mapping relationship between the foot and the leg inertial information. Synchronously the network can identify the gait characteristic phases of the walker. In other words, the input of the neural network model is six-axial inertial information of the leg, and the output is six-axial inertial information of the virtual foot IMU and the gait characteristic phases. To meet the positioning requirement without GNSS, the output of the VIMU is used to calculate attitude, velocity and position information through a strapdown navigation algorithm, and the detection conditions of gait phases (including the zero-velocity phase) with multiple constraints are designed according to the gait feature of the pedestrian. Furthermore, the error state equations and observation equations are established, with the velocity of the system taken as the observation. When the zero-velocity state or certain gait characteristic phases of pedestrian is detected, Kalman filter is triggered to estimate the error states and to compensate the navigation parameters of the pedestrian navigation system. In addition, since the headings of the foot and leg are basically the same according to the human lower limb kinematics model [23] the heading of human walking can be determined by the magnetic sensor information collected by a leg MIMU. The method proposed in this paper is suited to a clean magnetic environment without obvious interference from electromagnetic devices or magnetic materials, and the modification for magnetic interference will not be discussed further. The details of the pedestrian navigation system will be illustrated in Section 4.

## 3. Construction of VIMU Based on Machine Learning

Several problems exist when using a MEMS IMU to monitor pedestrian motion. Firstly, due to the influence of human physiological structure characteristics, though the extremities (such as foot) are necessary for monitoring the human motion, it is difficult to balance the accuracy and measurement range of the inertial sensors. Secondly, the distributed installation of MEMS IMUs will increase the hardware complexity of the pedestrian navigation system and decrease the reliability [23]. Aiming at addressing these issues, this section will put forward a method of building a virtual IMU based on machine learning, which can improve the accuracy of the inertial information, and widen the measurement range simultaneously.

### 3.1. Human Lower Limb Kinematics Model

The relationship between the motion information of each part of a pedestrian is usually described by the rigid body model under the premise of ignoring the flexible characteristics of human, as shown in Figure 2.

A fixed coordinate system and generalized coordinates are defined in Figure 2. In the fixed coordinate system, the ankle joint is taken as the origin and the radial direction is taken as the X-axis direction. Suppose that the vertical direction is the Y-axis direction and the Z-axis direction is determined by the right hand rule when the pedestrian moves with a certain heading angle. The generalized coordinates $\theta_{i}(i=1,\cdots ,5)$ and $\gamma_{i}(i=1,\cdots ,5)$ represent the angle between the linkages in the XY plane and the YZ plane, and the vertical direction and clockwise are defined as positive.

Thus the posture of the pedestrian can be uniquely determined by the origin of the fixed coordinate system and the generalized coordinates. As shown in Figure 2, $X_{h},Y_{h},Z_{h}$ are the coordinates of human hip joint; $X_{e},Y_{e},Z_{e}$ are the coordinates of the end of the swinging shank in the fixed coordinate system; $X_{b},Y_{b},Z_{b}$ are the coordinates of the ankle joint in the generalized coordinate system; $l_{4}$ and $l_{5}$ are the length of shank and thigh of the swinging leg, respectively; $d_{4}$ and $d_{5}$ are the absolute distance of the inertial sensor mounted on thigh and shank from the Y-axis direction; the marks A, B, and C represent the inertial device mounted on the thigh, the foot, and the shank respectively. For the remaining definitions readers can refer to [24]. Based on the forward kinematics theory, the relative positional relationship between the thigh and the foot, is shown in Figure 2:(1)$$x_{e}=x_{m_{4}}+\left(l_{4}-d_{4}\right)\cos\gamma_{4}\sin\theta_{4}+l_{5}\cos\gamma_{5}\sin\theta_{5}+l_{6}\cos\gamma_{6}\sin\theta_{6}$$
(2)$$y_{e}=y_{m_{4}}+\left(l_{4}-d_{4}\right)\cos\gamma_{4}\cos\theta_{4}+l_{5}\cos\gamma_{5}\cos\theta_{5}+l_{6}\cos\gamma_{6}\cos\theta_{6}$$
(3)$$z_{e}=z_{m_{4}}+\left(l_{4}-d_{4}\right)\sin\gamma_{4}+l_{5}\sin\gamma_{5}+l_{6}\sin\gamma_{6}$$

In the scheme proposed in this paper, the inertial information of position A (leg) and B (foot) approximately conforms to the rigid kinematic model. However, due to the non-rigid body characteristics of human body, it is difficult to establish an accurate model, so the relationship between the inertial information of position A and B cannot be obtained accurately. Therefore, a machine learning method is considered to construct the nonlinear model between the leg and the foot.

### 3.2. VGG-LSTM Hybrid Model Architecture

In order to construct the mapping model between the inertial information from the ipsilateral leg and foot, this paper studies and applies an improved deep hybrid network model, which is composed of one-dimensional serial convolutional neural network (visual geometry group network, or VGG network) and long short term memory network (LSTM network). The structure of the model is shown in Figure 3. The input of the VGG-LSTM hybrid network model is the six-axial inertial information of the leg, and the output is the six-axial inertial information of the ipsilateral foot, as well as the characteristic phases of the foot. Through sufficient training, the information of the leg IMU can be used to construct the virtual foot IMU in real time.

The convolutional neural model is connected serially instead of being set with a large convolutional core, so as to increase the depth of the model and provide more complex nonlinear transformation to extract higher dimensional features. The maximum pooling layer is used to reduce feature dimensions and control the risk of over-fitting while maintaining translation invariance [25]. On this basis, LSTM network receives the feature fragments of convolutional neural network, and further mines the time-order characteristics of inertial data via LSTM network, thus realizing the effect of maintaining long-term memory on the basis of short-term memory [26].

The convolutional neural network model includes the input layer $x^{0}$, convolutional layer c and pooling layer p. Generally, the input layer is set to layer 0, it can be modeled as:(4)$$x^{0}=\left[x_{1}^{0},\cdots ,x_{M}^{0}\right]$$
where *M* represents the size of the time window after data preprocessing. The output of the convolutional layer is:(5)$$\begin{array}{l} c_{i}=f(b+<q,x_{i}^{0},\cdots ,x_{i+\phi -1}^{0}>),\\ \;\;\;\;\;\;\;\;\;\;\;\;\;\;\;\;\;\;i=1,\cdots ,M-\phi +1 \end{array}$$
where f(⋅) is the activation function; b is the bias item; q is the one-dimensional convolutional kernel vector; ϕ is the length of q. The output of the pooling layer is shown in Equation (6):(6)$$\begin{array}{l} p_{j}=\max([c_{(j-1)R+1},\cdots ,c_{jR}]),\\ \;\;\;\;\;\;\;\;\;\;\;\;\;\;\;\;\;j=1,\cdots ,M/R \end{array}$$
where *R* represents the size of a pool window. The output of the pooling layer is the characteristic graph *p* learned by the convolutional kernel in the convolutional network, and multicore convolution means that each convolutional kernel *e* will generate a feature map *p^e^* in the convolutional process.

As shown in Figure 3, the LSTM network is an improved structure of the Recurrent Neural Network (RNN), which maintains long-term memory on the basis of adding input gate, output gate and forget gate to realize short-term memory. The one-dimensional feature maps output from the convolutional neural network are pieced together to form a one-dimensional eigenvector. The eigenvector s can be modeled as:(7)$$s=[p^{1},\cdots ,p^{n}]$$
where n represents the number of convolutional kernels. The eigenvector s is processed in the full connection layer, and the output of the full connection layer is;
(8)h=f(Ws+ε)
where W is the weight matrix of the full connection layer, ε is the connection layer offset item vector. The output of LSTM network is sent to the full connection layer of N nodes in turn, where N denotes the number of categories of motion.

In order to determine the optimal convolutional block number and layer number of LSTM network, we train the neural network models of different convolutional block numbers and layer numbers of LSTM network respectively. Then the optimal network structure is selected according to the statistics of CPU/GPU utilization rate and training performance. Finally, we found that when the convolutional block number is 4 and layer number of LSTM network is 3, the ratio of performance to power consumption reaches a minimum. Further increases of the complexity of the network model, will no longer significantly improve the approximation accuracy of VIMU. Therefore, the network structure of four convolutional blocks and three LSTM network layers can minimize the ratio between the approximation precision of VIMU and the system resource occupancy, that is, the system performance is optimal.

### 3.3. Construction of Training Model Based on VGG-LSTM Neural Network

The MTI-300 IMU (Xsense,) which is shown in Figure 4, consists of a tri-axial accelerometer, a tri-axial gyroscope, a tri-axial magnetometer, a 16-bit AD, internal DSP, nonvolatile memory and a serial transceiver, where the range of accelerometer is ±5 g, the range of gyroscope is ±600 °/s and the angular resolution is 0.05°. The main function of MTI-300 IMU is to collect the motion information data in real time, which will be transmitted for the construction of VGG-LSTM neural network model and pedestrian navigation calculation.

In order to build the training model based on VGG-LSTM neural network, two MTI-300 IMUs are installed respectively on the leg and foot of the same side, with the inertial information being collected synchronously. It can be seen from Figure 5 that the tri-axial information output of accelerometer and gyroscope varies when the gait changes at different stages. If the original data is taken as the network training data directly, the normalization degree of various parameters is poor, leading to the network performance deviation. Therefore, it is necessary to normalize the original sensor information output before it is used as the training samples of the VGG-LSTM neural network.

In this paper, the method of maximum value normalization is adopted, and the linear function normalizes the original data to the range of [0–1]. The normalization formula is shown as Equation (9):(9)$$X_{\mathrm{norm}}=\frac{X-X_{\min}}{X_{\max}-X_{\min}}$$

This method realizes equal scaling of the original data. In Equation (9), $X_{\mathrm{norm}}$ is the normalized data, X is the original data, with $X_{\max}$ and $X_{\min}$ denoting the maximum and minimum value of the original data set, respectively.

The output benchmark for the accuracy test of the VGG-LSTM neural network is the expected tri-axial gyroscope and accelerometer information, as well as the gait characteristic phases of pedestrian foot. As is shown in Figure 5, the expected output information of neural network includes $\omega_{\mathrm{xo}},\omega_{\mathrm{yo}},\omega_{\mathrm{zo}},a_{\mathrm{xo}},a_{\mathrm{yo}},a_{\mathrm{zo}}$. For the construction of the training model, the MEMS IMU information on pedestrian thigh, including $\omega_{\mathrm{xi}},\omega_{\mathrm{yi}},\omega_{\mathrm{zi}},a_{\mathrm{xi}},a_{\mathrm{yi}},a_{\mathrm{zi}}$, is taken as the input samples of the VGG-LSTM neural network, which is shown in Figure 6.

The typical motion of horizontal walking is taken as an example for illustration. The distributed structure inertial sensing system mentioned above is adopted to collect the inertial information of horizontal walking from one of the pedestrian lower limbs. The MEMS IMU information samples of the thigh position are taken as the training input of the VGG-LSTM neural network, and that of the foot position are taken as the training output and accuracy test benchmark.

Comparing VIMU information samples of the foot output by the network with the actual information samples collected by the Xsense MTI-300 in Figure 7 and Figure 8, the VGG-LSTM network can efficiently approximate the actual inertial information, and the values and periods keep persistently consistent during the horizontal walking process.

## 4. Pedestrian Navigation Algorithm

### 4.1. Pedestrian Navigation Algorithm

Real-time and accurate zero-velocity detection algorithm provides an important guarantee for error correction of foot inertial navigation system. Based on the virtual foot IMU, the periodic characteristics of accelerometer and gyroscope output and the foot characteristic phases can be utilized for the zero-velocity detection of multi-condition constraints. The detection algorithm is composed of two constraints, which are recorded as $\xi_{1}$ and $\xi_{2}$, respectively. The constraints are implemented as follows:

#### 4.1.1. Judge the Zero-Velocity State According to the Accelerometer and Gyroscope Output of the Virtual Foot IMU

Based on the virtual foot IMU, steps of automatically detecting zero-velocity state using original signals are described below:(10)$$\left\{ \begin{array}{l} a_{k}=a_{kx}^{2}+a_{ky}^{2}+a_{kz}^{2}\\ {\bar{a}}_{k}=\frac{1}{n+1}\sum_{j=k}^{k+n}a_{j}\\ \sigma_{ak}=\frac{1}{n+1}\sum_{i=k}^{k+n}{\left(a_{i}-{\bar{a}}_{k}\right)}^{2}\\ \lambda_{1,2}(k)=\left\{ \begin{array}{l} 1\;\;\;\;\;\;\sigma_{ak}<\varepsilon_{a_{1}}\;\\ 0\;\;\;\;\;\mathrm{others} \end{array}\right. \end{array}\right.$$
(11)$$\left\{ \begin{array}{l} a_{k}=a_{x}^{2}+a_{y}^{2}+a_{z}^{2}\\ \lambda_{3,4}(k)=\left\{ \begin{array}{l} 1\;\;\;\;\;\;a_{k}<\varepsilon_{a_{2}}\;\\ 0\;\;\;\;\;\;\mathrm{others} \end{array}\right. \end{array}\right.$$

In Equations (10) and (11), $a_{kx}$, $a_{ky}$, $a_{kz}$ represent the tri-axial acceleration or angular velocity collected by accelerometers or gyroscopes of virtual foot IMU respectively at moment *k*. n is the variance of the interval size; $\varepsilon_{a_{1}}$, $\varepsilon_{a_{2}}$ are the thresholds set according to the sensor accuracy of the VIMU. $\lambda_{1}$, $\lambda_{2}$, $\lambda_{3}$, $\lambda_{4}$ represent the detection results of zero-velocity interval under four different discriminant methods, respectively.

Under the condition of fixed parameter threshold, for different pedestrians and different walking paces, if only one of $\lambda_{1}$, $\lambda_{2}$, $\lambda_{3}$, $\lambda_{4}$ is used for zero-velocity discrimination, the stability of the judgment result will be poor, easily leading to misjudgment. In order to solve this problem, this paper proposes a new zero-velocity detection method, where four kinds of zero-velocity detection schemes are all adopted. The determination results of zero-velocity interval determined by the variance of accelerometers and gyroscopes ($\lambda_{1}$ and $\lambda_{2}$) are utilized to identify the initial time of zero velocity, and those determined by the amplitude of accelerometers and gyroscopes ($\lambda_{3}$ and $\lambda_{4}$) are utilized to identify terminal moment [27]. Each stage using the accelerometer and gyroscope at the same time improves the accuracy of zero-velocity judgment, and the joint determination procedure of zero velocity state is expounded as follows:(12)$$\xi_{1}=\left\{ \begin{array}{l} 1\;\;\;\;\;(\lambda_{1}(k)+\lambda_{3}(k)(\lambda_{2}(k)+\lambda_{4}(k))\geq 1\\ 0\;\;\;\;\;\;\;\;\;\;\;\;\;\;\;\;\;\;\;\;\;\;\;\;\mathrm{others} \end{array}\right.$$
where $\xi_{1}$ represents the comprehensive judgment program based on inertial information characteristics for stable gait (such as normal walking). It can accurately identify different pedestrians, different strides and different zero velocity intervals.

#### 4.1.2. Judge the Zero-Velocity State According to the Recognition of Zero-Velocity Phase from all Characteristic Phases of the Foot

Walking is a process of repeated movement of both feet, which is called gait cycle (GC). The typical pedestrian GC is demonstrated in Figure 9.

During walking, each GC contains a series of transition of typical postures and GC can also be divided into a series of time periods, which are called gait phases. Moreover, a walking cycle can be generally divided into supporting phase and swinging phase, or the more detailed classification consisting of 8 phases, which are generally expressed as the percentage of the whole GC [28]:(1)First touchdown period: the moment when the moving side heel touches the ground, accounting for about 2% of GC;(2)Load-bearing reaction period: it starts from the moment when the heel of the moving side touches the ground and lasts until the end when the toe of the opposite side leaves the ground. In the whole process, the sole of the moving side touches the ground completely, accounting for about 10% of GC;(3)In the middle stage of the supporting phase: from the moment when the toes on the opposite side are off the ground, and to the moment when the trunk is directly above the supporting leg, accounting for about 19% of GC;(4)The end of the supporting phase: form the moment when the supporting side heel leaves the ground to the moment when the opposite side heel follows the ground, accounting for about 19% of GC;(5)The earlier period of oscillation: from the moment when the opposite foot follows the ground to the moment before the toes on the supporting side leave the ground, accounting for about 12% of GC;(6)Early swing phase: from the moment when the foot is off the ground to the moment when the knee reaches the maximum bending state, accounting for about 13% of GC;(7)Middle swing phase: from the moment when the knee joint reaches the bending state to the moment when the calf swings to the place where it is perpendicular to the ground, accounting for about 12% of GC;(8)End of swinging phase: from the moment when the calf is perpendicular to the ground to the moment when the heel touches the ground again, accounting for about 13% of GC.

Then the gait phases recognition method is as follows: according to the analysis of pedestrian gait characteristics, the tri-axial virtual accelerometer and gyroscope information of each GC is divided into eight data samples according to the percentage of different gait phases. The data samples and corresponding gait phases are imported into the above VGG-LSTM hybrid model as input and output, respectively, for training, thus the network model can accurately identify the eight gait phases. Specially, when the neural network identifies the zero-velocity phase shown in Figure 9, the zero-velocity phase can be determined as follows:(13)$$\xi_{2}=\left\{ \begin{array}{l} 1\;\;\;\;\;\;\mathrm{zero}\text{-}\mathrm{velocity}\;\mathrm{phase}\\ 0\;\;\;\;\;\;\mathrm{others} \end{array}\right.$$
where $\xi_{2}$ represents the zero-velocity discriminant result at the moment k. In order to avoid the problem that a single detection algorithm cannot accurately determine a pedestrian’s complex gait, the logical “and” operation is carried out between the two discriminant results $\xi_{1}$ and $\xi_{2}$, then ξ is used to represent the final zero-velocity detection result:(14)$$\xi =\xi_{1}\&\xi_{2}$$

#### 4.1.3. The Accuracy of Zero-Velocity Detection Algorithm

In order to verify the detection accuracy of the zero-velocity interval detection algorithm, three groups of data are collected in the gaits of horizontal walking, upstairs, downstairs, upslope, downslope and fast walking. For each group, 100 steps are taken for each gait. The reference for the zero-velocity interval is captured by the high-precision visual system simultaneously. The formula of zero-velocity interval detection accuracy is:(15)$$Acc=1-\frac{\left|T_{\mathrm{test}}-T_{\mathrm{true}}\right|}{T_{\mathrm{true}}}$$
here $T_{\mathrm{test}}$ is the period of detected zero-velocity interval, and $T_{\mathrm{true}}$ is the period of actual zero-velocity interval captured by visual system. The proposed algorithm is compared with double threshold zero-velocity interval detection algorithm [29] and adaptive threshold zero-velocity interval detection algorithm [30] the comparison results are shown in the Table 1.

The number of walking steps can be also detected by zero-velocity intervals, the accuracy of step number detection is shown in Table 2.

### 4.2. Design of the Kalman Filter

The process of using ZUPT to assist SINS can be divided into two states: (1) Non-zero velocity state: the system cannot obtain the velocity error observation, and Kalman filter only updates partly; (2) Zero velocity state: the velocity error observations are obtained, Kalman filter fully updates, and the estimated value of system error is fed back and compensates SINS [31]. The Kalman filter based on the virtual foot inertial navigation system is as follows:

#### 4.2.1. State Equation

The inertial navigation system error model consists of inertial navigation platform error angle model, velocity error model, position error model and inertial instrument error model.

Inertial navigation platform error angle equation:(16)$$\dot{\varphi }{=\delta \omega }_{\mathrm{in}}^{n}+\varphi \times \omega_{\mathrm{in}}^{n}+\varepsilon^{n}$$
where E, N and U marked below the angles represent the east, north and upward directions of the geographic coordinate system respectively.

Velocity error equation:(17)$$\delta \dot{V}={\dot{V}}^{c}-V^{n}=f^{n}\times \varphi +\nabla^{n}-(2\delta \omega_{\mathrm{ie}}^{n}+\delta \omega_{\mathrm{en}}^{n})\times V^{n}\;-(2\omega_{\mathrm{ie}}^{n}+\omega_{\mathrm{en}}^{n})\times \delta V+\delta g$$

Position error equation:(18)$$\left\{ \begin{array}{l} \delta \dot{L}=\frac{\delta v_{N}}{R_{M}+h}\\ \delta \dot{\lambda }=\frac{\delta v_{E}}{R_{N}+h}\sec L+\frac{v_{E}}{R_{N}+h}\sec L\tan L\delta L\\ \delta \dot{h}=\delta v_{U} \end{array}\right.$$

Random constant error model of inertial devices:(19)$$\dot{\varepsilon }=0$$
(20)$$\dot{\nabla }=0$$

Therefore, the 15-dimensional state equation of the system can be expressed as Equation (21):(21)$$\dot{X}(t)=F(t)X(t)+G(t)W(t)$$
where F is the state coefficient matrix, G is the error coefficient matrix, W is the system white noise vector, X is the error state vector, which is given by Equation (22):(22)$$X={\left[\begin{array}{cccc} \varphi_{E} & \varphi_{N} & \varphi_{U} & \begin{array}{ccc} \delta v_{E} & \delta v_{N} & \begin{array}{ccc} \delta v_{U} & \delta L & \begin{array}{ccc} \delta \lambda & \delta h & \begin{array}{ccc} \varepsilon_{x} & \varepsilon_{y} & \begin{array}{ccc} \varepsilon_{z} & \nabla_{x} & \begin{array}{cc} \nabla_{y} & \nabla_{z} \end{array} \end{array} \end{array} \end{array} \end{array} \end{array} \end{array}\right]}^{T}$$

#### 4.2.2. Observation Equation

The velocity observation equation is shown in Equation (23):(23)$$Z_{v}(t)=\left[\begin{array}{l} v_{\mathrm{IN}}-0\\ v_{\mathrm{IE}}-0\\ v_{\mathrm{IU}}-0 \end{array}\right]=H_{v}(t)X(t)+V_{v}(t)$$
where $V_{v}$ is the observation white noise vector, $H_{v}$ is the observation matrix, which is shown as follows:(24)$$H_{v}=\left[0_{3*3}\vdots diag\left[\begin{array}{ccc} 1 & 1 & 1 \end{array}\right]\vdots 0_{3*9}\right]$$

The ZUPT and Attitude Matching modes of the pedestrian navigation system can be adjusted according to the available observed data:Non-zero velocity state: no error observation, Kalman filter only updates partly, the system has no feedback of error estimation;Zero-velocity state: both the velocity and the attitude error are available, and the error observation is:
(25)$$Z(t)=\left[Z_{v}(t)\right]$$

The specific meaning of each physical quantity in Equations (16)–(25) and the process of discretization of the system model are shown in [32].

## 5. Experiment

To verify the feasibility of the proposed method in practical application, Xsens MTI-300 IMUs are installed on the left thigh and the back of the left foot, respectively. A portable computer is used for real-time data collection and navigation algorithm calculation, and thus a prototype of the pedestrian navigation system can be formed. 

Indoor and outdoor navigation experiments are carried out to analyze the positioning performance and error characteristics. The indoor routine is about 240 m, while the outdoor routine is about 360 m. In the first group of experiment, the pedestrian walks in the conventional walking pace with a velocity of about 1.0 m/s. The total indoor walking time is about 240 s, and the total outdoor walking time is about 360 s accordingly. The second group of experiment adopts the fast walking pace with a velocity of about 2.5 m/s, and the indoor walking time is about 100 s, and the outdoor walking time is about 140 s accordingly. Figure 10a,b show the virtual and actual IMU gyroscope X-axial output and accelerometer Y-axial output under normal gait conditions of 1 m/s, respectively, and Figure 10c,d are virtual and actual IMU gyroscope ***X***-axis output and accelerometer ***Y***-axis output under fast gait conditions of 2.5 m/s, respectively. The performance comparison of the pedestrian navigation method is shown in Figure 11.

Figure 11a,c are the pedestrian navigation experiment routines in an indoor and outdoor environment, respectively. 

Figure 11b,d are the performance comparison curves of indoor and outdoor pedestrian navigation respectively, including:(1)Curve (a) is calculated from actual foot IMU at a conventional walking pace, the navigation method is assisted by the ZUPT algorithm, and the indoor positioning error of the scheme is 1.5 m, accounting for about 0.6% of the total length of the walking distance; the outdoor positioning error is 2.4 m, accounting for about 1.0% of the total length;(2)Curve (b) is calculated from actual foot IMU at a fast walking pace, and the navigation method is assisted by ZUPT algorithm. In the later stage of walking, the extremum of angular velocity of human foot is greater than 600 (°)/s, and the extremum of acceleration is greater than 10 *g*. Therefore, in the case of over-range of IMU, neither indoor nor outdoor conventional ZUPT algorithm assisting methods can achieve better navigation performance;(3)Curve (c) is calculated from a virtual foot IMU built on the lower limb at a fast walking pace and the navigation method adopted is assisted by the ZUPT algorithm. It can be seen from the curves that the method proposed in this paper can realize navigation and positioning function at a fast walking pace, and will not be significantly affected by the over-range of IMU. However, there are training errors in the neural network model. To be specific, the indoor positioning error of the scheme is 4.8 m, accounting for about 1.3% of the total length of the walking distance; the outdoor positioning error is 7.6 m, accounting for about 2.1% of the total length of the walking distance.

It can be concluded from the comparative analysis of the above experimental results that high-precision navigation and positioning function can be achieved by using the actual foot IMUs combined with a ZUPT algorithm under conventional walking pace conditions.

In the case of fast walking pace, the movement of human lower limb is close to or even beyond the range of the sensor component, and the lower limb may also experience shock, vibration and other phenomena, which makes it impossible to effectively realize the ZUPT algorithm and hence navigation and positioning. In contrast, in terms of fast walking pace, the navigation and positioning method based on virtual foot IMU and ZUPT algorithm can achieve relatively accurate pedestrian navigation and positioning.

## 6. Conclusions

This paper proposed a novel navigation method based on the construction of a virtual foot IMU and pedestrian gait feature assistance. In this method, based on the VGG-LSTM neural network model, the nonlinear mapping relationship between inertial information from human foot and leg is established and then the construction of virtual foot IMU and VINS is realized. On the basis of the periodic characteristics of accelerometer and gyroscope output of virtual foot IMU, as well as the judgment of foot characteristic phases in the output of neural network model, the zero-velocity detection with multiple conditions is carried out. The experimental results show that the integrated method of error modifying proposed in the paper can effectively slow down the accumulation of positioning error in the gait types that exceed inertial sensors measuring range.

## Figures and Tables

**Figure 1 sensors-20-01530-f001:**
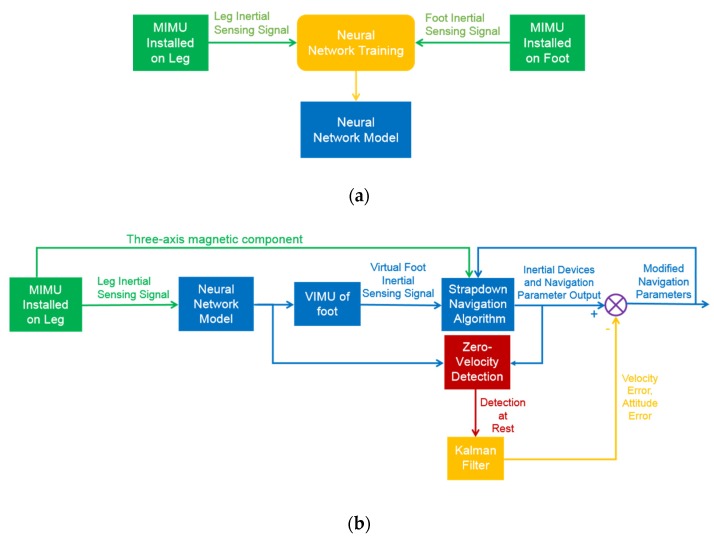
The overall scheme of pedestrian navigation method (**a**) Construction of neural network; (**b**) Diagram of the pedestrian navigation process.

**Figure 2 sensors-20-01530-f002:**
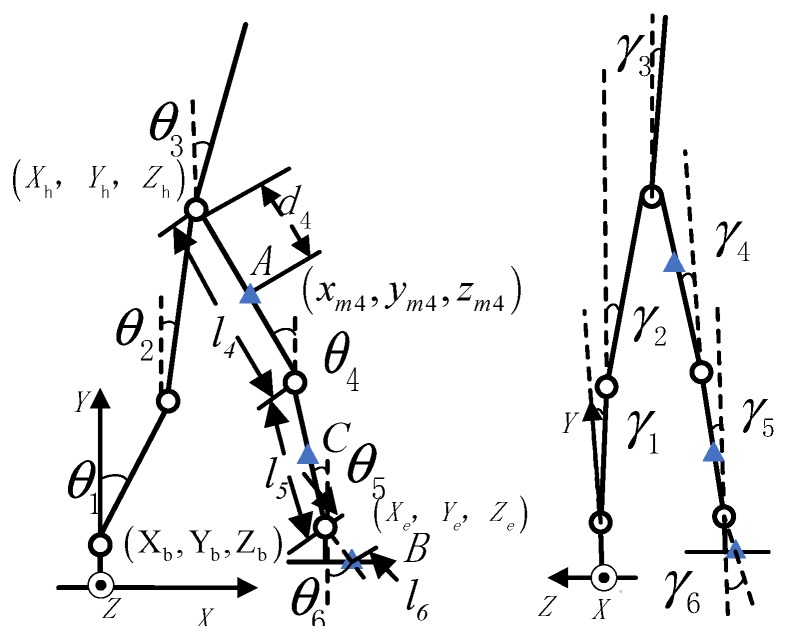
Side view and front view of human lower rigid-body kinematics model.

**Figure 3 sensors-20-01530-f003:**
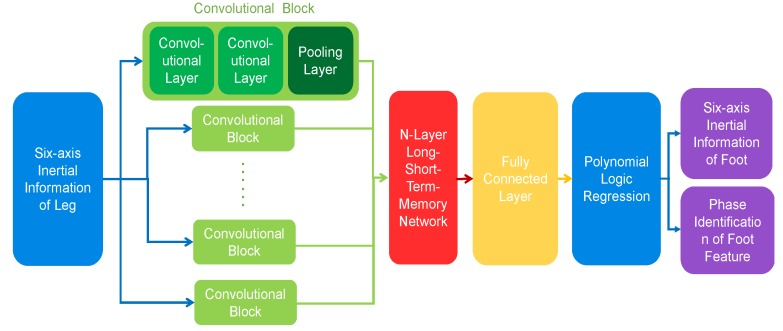
VGG-LSTM model architecture.

**Figure 4 sensors-20-01530-f004:**
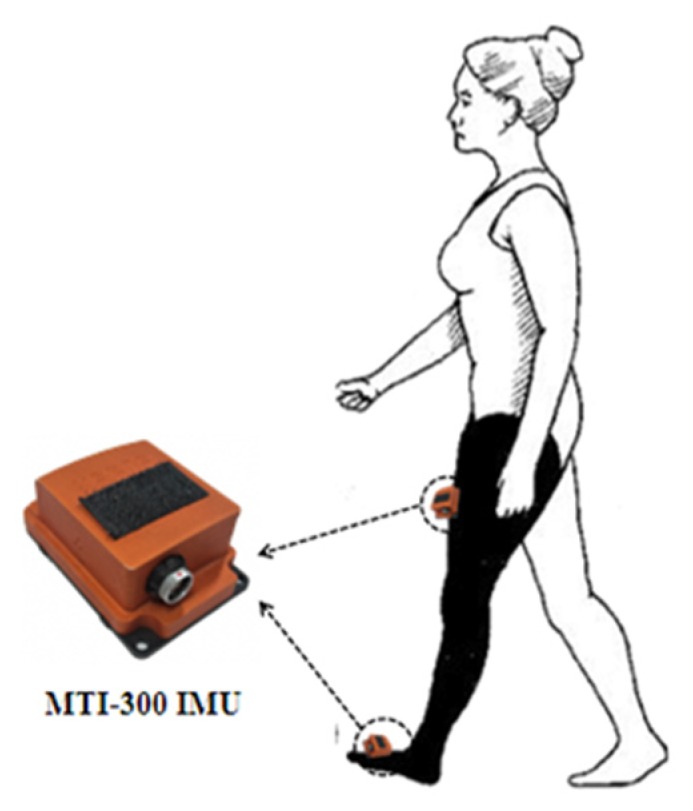
Distributed structure for inertial information acquisition.

**Figure 5 sensors-20-01530-f005:**
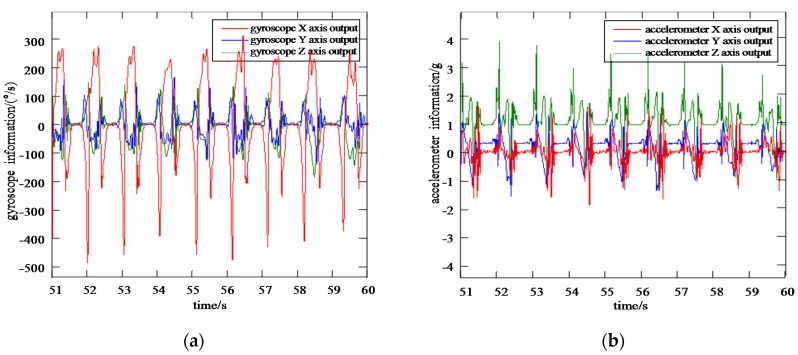
Gyroscope and accelerometer information from pedestrian foot (**a**) Gyroscope information from pedestrian foot; (**b**) Accelerometer information from pedestrian foot.

**Figure 6 sensors-20-01530-f006:**
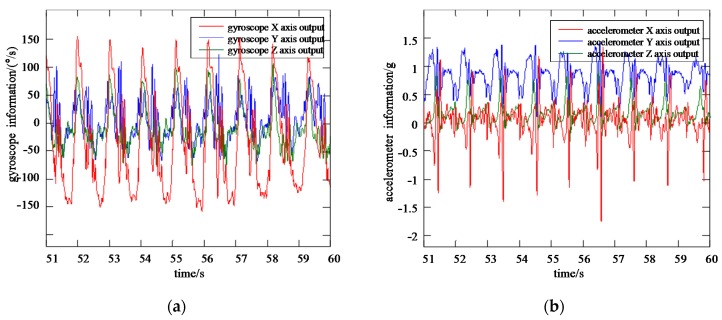
Gyroscope and accelerometer information from pedestrian thigh (**a**) Gyroscope information from pedestrian thigh; (**b**) Accelerometer information from pedestrian thigh

**Figure 7 sensors-20-01530-f007:**
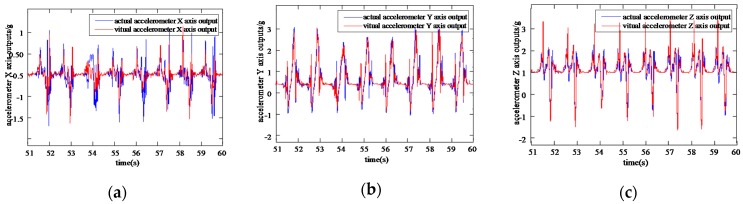
Information comparison of virtual accelerometers with reference accelerometers (**a**) Comparison of virtual accelerometer X axis information with reference accelerometer X axis information; (**b**) Comparison of virtual accelerometer Y axis information with reference accelerometer Y axis information; (**c**) Comparison of virtual accelerometer Z axis information with reference accelerometer Z axis information.

**Figure 8 sensors-20-01530-f008:**
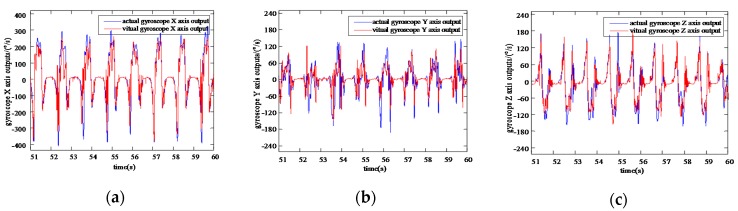
Information comparison of virtual gyroscope with reference gyroscope (**a**) Comparison of virtual gyroscope X axis information with reference gyroscope X axis information; (**b**) Comparison of virtual gyroscope Y axis information with reference gyroscope Y axis information; (**c**) Comparison of virtual gyroscope Z axis information with reference gyroscope Z axis information.

**Figure 9 sensors-20-01530-f009:**
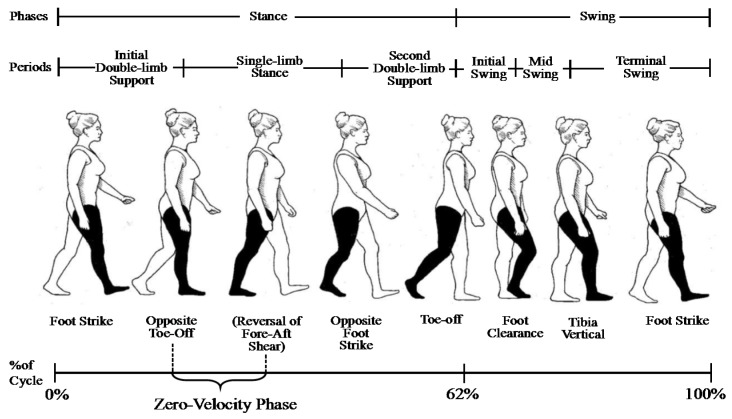
Normal gait cycle of pedestrian.

**Figure 10 sensors-20-01530-f010:**
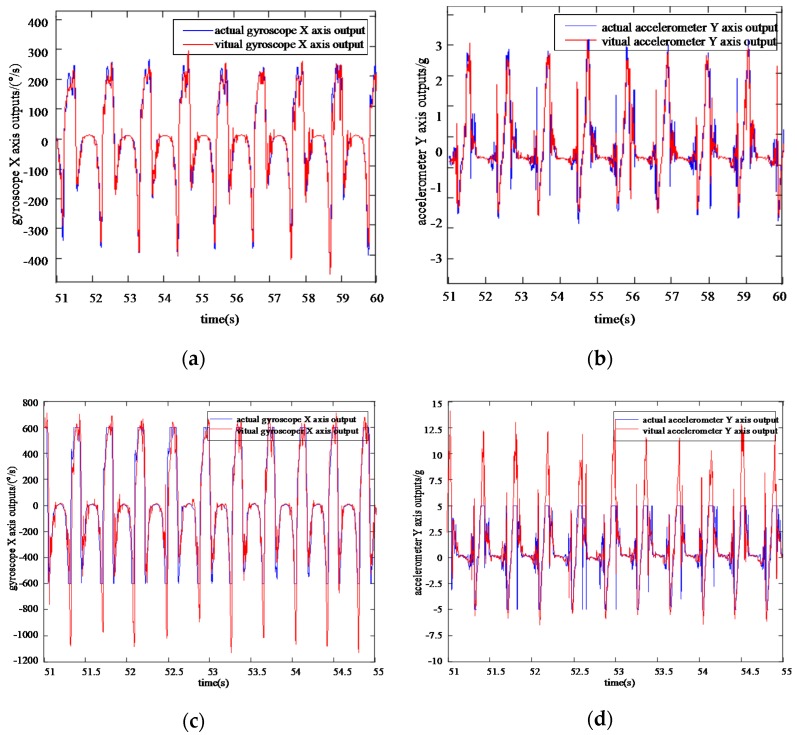
Virtual and actual IMU data under the walking velocity of 1 and 2.5 m/s: (**a**) X axis virtual and actual gyroscope outputs with a walking velocity of 1 m/s; (**b**) Y axis virtual and actual accelerometer outputs with a walking velocity of 1 m/s; (**c**) X axis virtual and actual gyroscope outputs with a walking velocity of 2.5 m/s; (**d**) Y axis virtual and actual accelerometer outputs with a walking velocity of 2.5 m/s.

**Figure 11 sensors-20-01530-f011:**
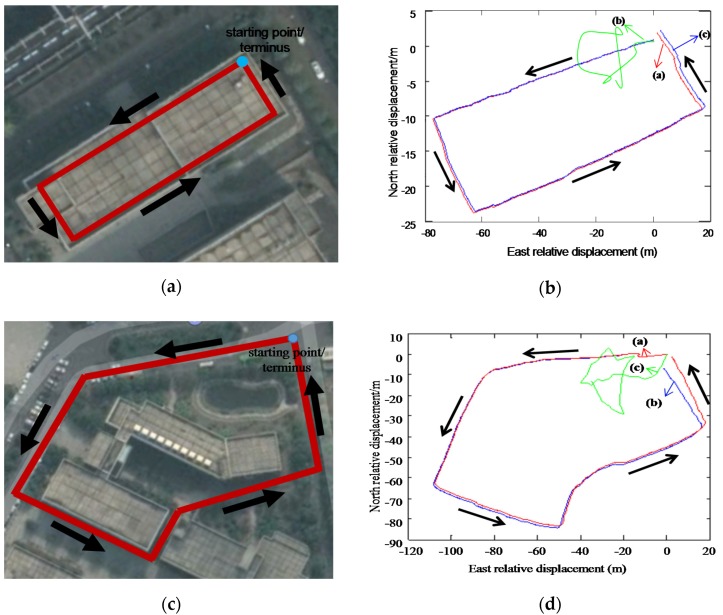
Performance verification experiments of pedestrian navigation methods: (**a**) Indoor pedestrian navigation experiment route; (**b**) Comparison of indoor curves by different pedestrian navigation methods; (**c**) Outdoor pedestrian navigation experiment route; (**d**) Comparison of outdoor curves by different pedestrian navigation methods.

**Table 1 sensors-20-01530-t001:** The comparison results of zero-velocity interval detection accuracy.

Gait Types	The Zero-Velocity Interval Detection Accuracy		
	Double Threshold Algorithm	Adaptive Threshold Algorithm	Proposed Algorithm
Horizontal walking	98.7%	98.9%	99.3%
Upstairs	98.3%	98.5%	98.8%
Downstairs	98.2%	98.3%	98.7%
Upslope	98.5%	98.7%	99.1%
Downslope	98.3%	98.6%	99.0%
Fast walking	97.6%	98.0%	98.5%

**Table 2 sensors-20-01530-t002:** The comparison results of zero-velocity interval detection accuracy.

Gait Types	Actual Step Numbers	Detected Step Numbers	The Accuracy of Detection
Horizontal walking	500	500	100.0%
Upstairs	500	500	100.0%
Downstairs	500	500	100.0%
Upslope	500	500	100.0%
Downslope	500	500	100.0%
Fast walking	500	496	99.2%

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
