# Peer review of "Pedestrian Navigation Method Based on Machine Learning and Gait Feature Assistance"

_sensors, 2020, doi:10.3390/s20051530_

Round 1

Reviewer 1 Report

There are some correlations between the movements of various units of human body. In order to overcome the measurement error of the wearable MIMU installed on foot, this paper gives a method based on construction of Virtual Inertial Measurement Unit (VIMU), and the method has been validated by corresponding experiments. The paper’s idea is novel.

The specific problems as follows:

Figure 1 shows that tri-axial magnetic component is used to correct the magnetic heading. The environmental limitations of magnetic sensor applications as well as the magnetic interference condition of this paper should be explained more clearly. A figure should be given to reflect the relationship between the input and output variables and to illustrate the structure of the neural network. The physical meaning of the input and output variables should be explained. It's easy to establish a correlation between the foot and leg movement features. Because of the leg’s lever arm effect and nonrigid characteristic, there exist a few differences between foot’s and leg’s inertial navigation parameters.

In the process of pedestrian walking, the zero-velocity intervals of foot are very evident. Please specify how to solve the problem that the leg’s zero-velocity intervals don’t correspond to the foot’s zero-velocity intervals.

Reviewer 2 Report

This paper proposed a pedestrian system by collecting the inertial data from different positions of pedestrian’s lower limb synchronously with gait feature assistance. The idea is somewhat novel, but the following questions should be addressed before publication.

Please use gyroscope in the paper instead of gyroscoper.  

Why do you need two IMUs mounted on different positions, and why on legs and foots? The sentence of “In order to construct the neural network model to establish the nonlinear mapping relationship function between the inertial information of foot and leg, MEMS IMUs should be installed respectively on the leg and foot of the same side of lower limb, with the tri-axial acceleration and tri-axial angular velocity of the IMUs collected at the same frequency as the training samples of the neural network model” is not very clear.

Do the authors think the comprehensive data mining will be better in precision of the navigation than the data just collected from a single position like thigh?

The only data the authors provided in this paper is related to the real/virtual accelerometer and gyroscope, which is not very convince. It’s like that the author show us an algorithm black box and get the result. The authors must add more data in processing. For example, can you show the experimental results that your proposed zero detection method is more precise than the conventional one? Can you also give the data of the step detection error using your in comparison with the real steps?

How do you process with different step modes such as walking, running, up/down stairs?
